# Preference learning for guiding the tree search in continuous POMDPs

**Jiyong Ahn, Sanghyeon Son, Dongryung Lee, Jisu Han, Dongwon Son, and Beomjoon Kim**
Graduate School of AI at KAIST
{ajy8456, ssh98son, dlee960504, jshan, dongwon.son, beomjoon.kim}@kaist.ac.kr

**Abstract:** A robot operating in a partially observable environment must perform sensing actions to achieve a goal, such as clearing the objects in front of a shelf to better localize a target object at the back, and estimate its shape for grasping. A POMDP is a principled framework for enabling robots to perform such information-gathering actions. Unfortunately, while robot manipulation domains involve high-dimensional and continuous observation and action spaces, most POMDP solvers are limited to discrete spaces. Recently, POMCPOW [1] has been proposed for continuous POMDPs, which handles continuity using sampling and progressive widening [2]. However, for robot manipulation problems involving camera observations and multiple objects, POMCPOW is too slow to be practical. We take inspiration from the recent work in learning to guide task and motion planning [3] to propose a framework that learns to guide POMCPOW from past planning experience. Our method uses preference learning [4, 5, 6, 7] that utilizes both success and failure trajectories, where the preference label is given by the results of the tree search. We demonstrate the efficacy of our framework in several continuous partially observable robotics domains, including real-world manipulation, where our framework explicitly reasons about the uncertainty in off-the-shelf segmentation and pose estimation algorithms. Details of the project are accessible in the following URL: `https://sites.google.com/view/preference-guided-pomcpow?usp=sharing`.

**Keywords:** POMDP, Online planning, Guided Search, Preference-based learning

## 1 Introduction

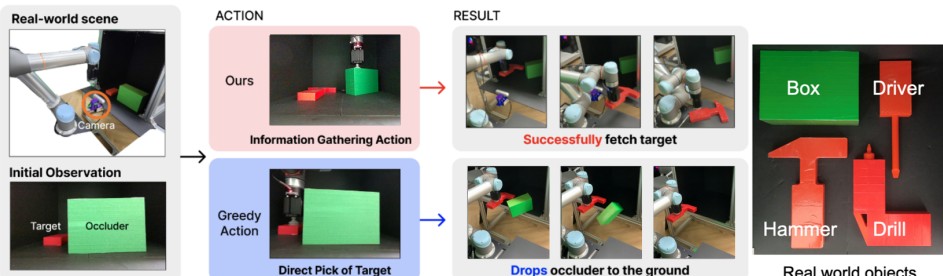

Figure 1: An object fetching task from a tight cabinet. The camera is at a location marked with orange, and the initial observation is shown at the bottom left. The target can be any one of the red objects from the right-most figure, and the occluder is the green box. The occluder prevents accurately estimating the target object's shape and pose. Only by performing information-gathering actions can we safely fetch the target object.

Consider a robot manipulator tasked with retrieving a target object occluded by another object in a tight cabinet. Due to significant occlusion, the exact shape and location of the target object are

7th Conference on Robot Learning (CoRL 2023), Atlanta, USA.

unknown, and rashly fetching the object without gathering these pieces of information could make the occluding object fall. Our goal is to enable robots to efficiently compute plans that consider these uncertainties and perform information-gathering actions to robustly operate in unstructured environments. This is a challenging problem that involves partial observability in high dimensional continuous state, action, and observation spaces. Figure 1 demonstrates an example scenario.

Typically, such problems are modeled with POMDPs, but the primary difficulty in using POMDP solvers is their computational complexity. To resolve this, several sampling-based online planning algorithms, such as POMCP [8, 1], have been proposed, which effectively handle large-scale discrete POMDPs by using Monte Carlo Tree Search (MCTS). Instead of computing the solutions for *all* belief states [9, 10, 11, 12, 13, 14, 15], POMCP performs a tree search to compute the solution for a single initial belief state. A recent extension, Partially Observable Monte Carlo Planning with Observation Widening (POMCPOW) [1], extends POMCP to continuous observation and action spaces using progressive widening [2] in both observation and action nodes. With this extension, POMCPOW can readily be applied to robotics problems involving continuous spaces.

However, even with POMCPOW, the computational challenge stemming from continuous and high-dimensional observations, actions, and states is still significant. In [3], the authors propose a method for learning to *guide* planning for challenging long-horizon task and motion planning problems in continuous spaces. The idea is to learn the value function and policy from past planning experience in order to speed up the tree search in a similar fashion to AlphaGo [16]. However, these methods are limited to fully observability environments and cannot handle state uncertainty.

In this work, we extend [3] to a partially observable setup. Like [3], we learn a value function and policy for guiding a tree search. However, unlike in that work, where the value function and policy operate on fully observable states, ours operate on action-observation histories and guides POMCPOW. In this context, the simplest way to train a value function would be to gather all history and future-sum-of-reward pairs from past search trees, and learn a mapping from history to its value. However, this approach would require an unthinkably large amount of data to train an effective value function due to the infinite possible histories containing high dimensional observations. As generating a dataset involves performing a tree search in a POMDP, such large-scale data generation would require a significant amount of time.

We propose an alternative data-efficient technique for learning a value function based on the following two observations. First, a search tree for a POMDP typically consists of a few success histories that led to a goal and a large number of other histories that did not. Second, all we need to efficiently guide a tree search is ranking among the histories specifying which one is more likely to lead to the goal, not their actual values, since the purpose of a value function in a tree search is to determine Based on these two observations, we propose a value function learning algorithm that learns the ranking among histories.

More concretely, We utilize the preference function instead of a regressor because it suffices to determine the exploration priority among nodes. In preference-based *reward* learning [4, 7, 5, 6], you are presented with two trajectories, and an oracle indicates the preferred one. Subsequently, a reward function is trained to assign greater rewards to the preferred trajectory compared to the other. We adapt this concept to value function learning, where preference labels are derived from the outcomes of tree searches. Within a search tree, we select a history that reached a goal, termed as a *success history*, and another that did not, termed as a *failure history*. We learn a value function that favors the success history over the failure history. One potential limitation of this straightforward success-and-failure preference labeling approach could be the absence of optimality consideration. To address this, we generate additional data by pairing two successful histories and labeling the one closer to the goal as the preferred one. We discovered that in scenarios with limited data, preference learning proves more robust than regression because it is less susceptible to variations in value differences, demonstrating greater resilience against noise in comparison to regression, which exhibits higher variance. Figure 2 for an illustration of our labeling scheme.

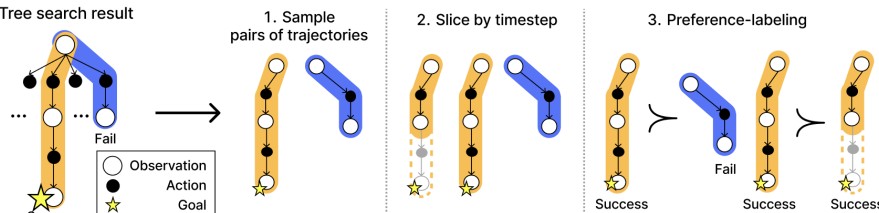

Figure 2: **Preference-dataset generation.** Given the tree search result, we first sample a pair of trajectories, at least one of which is a success trajectory. We then slice a subset of them at a particular time step, as shown on the left-most trajectory in step 2, and then label them with preferences as follows: if the pair consists of success and failure trajectories, we prefer the success trajectory (left in step 3). If the pair consists only of success trajectories, then we prefer the one that is closer to the goal (right in step 3).

For training a policy, we could, in principle, imitate the actions on success histories from past planning experiences using regression. However, to facilitate efficient exploration during tree search, we need a multi-modal policy rather than a single-modal policy because, with a single-mode policy, you are likely to only explore similar actions in a given node [17]. Furthermore, if we imitate success histories, then we are limited to a small number of histories per search tree, which is data inefficient. We instead propose to use an energy-based model, where the energy is based on a Q-function, which effectively defines a multi-modal policy whose probability of action is proportional to its Q-value [17]. We train the Q-function simultaneously with the value function by implementing these two functions as two heads that share the same backbone transformer but with different inputs. Preference-V, the value function trained with preference labels, consumes the sequence of action-observation history up to time step $t$, while Preference-Q does not see the last observation. Figure 3 shows the architecture of the preference value functions.

The problem with an energy-based function in continuous space is that exact sampling is intractable. One way to get around this is to uniform-randomly sample several actions, evaluate their values, and then choose the one with the highest Q-value. However, while fast, this procedure generates poor-quality actions, especially in high-dimensional spaces. Alternatively, we can use MCMC sampling [18], which, given a sufficient amount of time, can generate actions with high Q-values. However, the inference time is typically too long to be practical. Instead, we propose to use VAE [19] to approximate the Preference-Q-based energy function. At every gradient step, we uniform-randomly sample $N$ number of actions and minimize the Q-function-weighted loss for training the VAE, so that the higher Q-value, the higher the chance of being sampled from the VAE. This approach generates higher-quality samples than uniform random sampling because the VAE sees $N$ number of new actions at every gradient step, and has a much faster inference time than MCMC sampling because the inference is just a simple feed-forward prediction using a VAE.

We call our framework PGP (Preference-Guided POMCPOW). In our experiments, we demonstrate that PGP achieves a superior data efficiency than imitation-guided POMCPOW, and that it is more computationally efficient than unguided POMCPOW. In a real-world robot experiment, we demonstrate a system that integrates the off-the-shelf perception algorithms, POMCPOW, and our learning framework to solve the object fetching problem shown in Figure 1.

## 2 Related work

**Algorithms for POMDPs** To combat the curse of history, several online POMDP algorithms have been proposed. Unlike the conventional methods that compute solutions off-line for all possible belief states using full-width backup algorithms such as value iteration [9, 10, 11, 12, 13, 14, 15], online planning algorithms use tree search to compute a solution for a single belief state [8, 20]. In particular, POMCP [8] combines MCTS and unweighted particle filtering to apply a sampling-based online planning algorithm to solve large-scale discrete POMDP. However, for POMDPs with continuous action and observation spaces, POMCP cannot build a search tree for more than one step

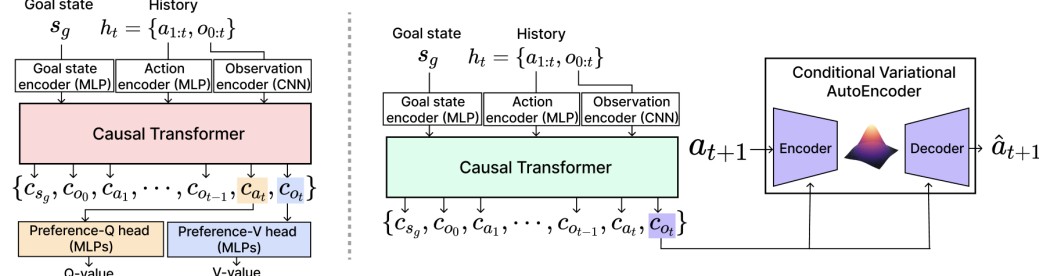

Figure 3: **The network architectures** (a) The value networks receive inputs of different types: goal state $s_g$, action $a_t$, and observation $o_t$. The encoders encode them into vectors of the same dimension. The causal transformer turns these encoded vectors into context vectors, $\{c_{s_g}, c_{a_{1:t}}, c_{o_{0:t}}\}$, where $c_{o_t}$ depends on $h_t$, while $c_{a_t}$ depends on $h_{t-1}a_t$, without $o_t$. The preference-Q head receives $c_{a_t}$ as input, and the preference-V head receives $c_{o_t}$. (b) The policy network has a similar architecture as the value networks, but uses a CVAE [19] that conditions on $c_{o_t}$.

because, for each new simulation, a new action and observation branch is expanded every time, as the probability of sampling the same action and observation from a continuous space is zero.

POMCPOW [1] extends POMCP to continuous spaces by using progressive widening (PW) [2] which samples a new branch only if the number of branches in that node is below $k \cdot M^\alpha$, where $k > 0$ and $\alpha \in (0, 1)$ are user-defined hyper-parameter and $M$ is the number of visits to that node. Otherwise, it revisits existing branches in order to improve their value estimates. POMCPOW uses PW both at action and observation nodes and integrates it with UCT [21] to handle the exploration-exploitation trade-offs in continuous spaces. The use of PW in POMCPOW specifies *when* we should sample a new action but does not specify *which* new action we should sample next. Further, it uses a random roll-out to estimate the value of a leaf node, which only gives a very crude estimate. PGP uses a policy for sampling a new action and value function to replace the roll-out in POMCPOW.

**Learning to guide tree search** The most prominent system for guiding a search is AlphaGo [22, 16, 23, 24]. AlphaGo guides the MCTS using a policy network that recommends actions and a value network that predicts the probability of winning at a given state, both of which are learned offline using self-play. Like AlphaGo, we learn a value function and policy to guide the search. One key difference is that for us, we learn from *planning experience*, while AlphaGo learns from *execution experience*, which is more expensive to obtain. Another key difference is that AlphaGo is limited to discrete action and state spaces, with full state information, whereas we are considering continuous state, observation, and action spaces with partial observability similar to [25].

The body of work most closely related to our framework includes algorithms for learning to guide task and motion planning based on planning experience [3, 26, 27, 28, 29, 30]. In [29, 30, 27], the authors propose a method for learning a visual heuristic function to guide the discrete task level search. In [26, 28], the authors propose strategies for learning a policy using variants of Generative Adversarial Networks to guide the continuous motion-level search. In [3], the authors integrate both the policy and the value function learned from planning experience with a tree search to solve long-horizon manipulation problems. Like us, they use planning experience to learn to guide the search in continuous state and action spaces. PGP is an extension of these works to a partially observable setup with state uncertainty.

**Learning from preference** Several works utilize *preference* from humans to train agents, by ranking the pairs of trajectory segments [7]. For instance, Large Language Models (LLMs) are fine-tuned to use human preferences as a reward signal [4], and reward functions are learned from preferences given by a human in games [5] and robotics [31, 32, 33]. Inverse reinforcement learning also learns a reward function from a sequence of ranked demonstrations [6]. The key difference from our work is that we are given a reward function, and the goal is to learn the value function that expresses the preference over different trajectories using the result of tree search as a labeler.

# 3   Learning the preference value and policy networks for PGP

Solving previous planning problems provides us with a set of search trees. There are largely three steps for using PGP: (1) generating the preference dataset from the search trees, (2) training the value and policy networks, and (3) deploying them to tree search. We provide the pseudocode for step (3) in the appendix. We first describe the data-generation process.

In POMCPOW, we simulate histories to find the solution to the given planning problem. So, solving a problem gives us a search tree containing a set of histories. For each problem, we assume that we have a set of goal states. Each history is defined as $h_t = \{o_0, a_1, o_1, \cdots, a_t, o_t\}$ where, $a_t$ and $o_t$ are action and observation at time step $t$ respectively. We label a history as *success* if the belief particle in POMCPOW that corresponds to that history got to the goal. We can identify that with certainty because a particle is a state sample from a belief state. Suppose we are given a pair of histories $(h_{t^i}^i, h_{t^j}^j)$, at least one of which is a success. Let $G^i$ indicate whether the history $h_{t^i}^i$ is part of a success history $h_{T^i}^i$, where $T^i$ is the length of the history to a termination and $t_i$ is the sampled time step in the history such that $T^i > t_i$. We label the pair of history, $(h_{t^i}^i, h_{t^j}^j)$, as

$$y(h_{t^i}^i, h_{t^j}^j) = \begin{cases} (1,0) & \text{if } G^i = 1 \text{ and } G^j = 0 \\ (0.5, 0.5) & \text{if } G^i = G^j = 1 \text{ and } T^i - t^i = T^j - t^j \\ (0,1) & \text{if } G^i = G^j = 1 \text{ and } T^i - t^i > T^j - t^j \end{cases} \tag{1}$$

In simple terms, if the $i^{th}$ history is part of a success history and the $j^{th}$ history is not, then we favor the $i^{th}$ history. If both are part of success histories and have an equal number of remaining time steps to the goal, they are both equally preferred. However, if both are success histories but the $j^{th}$ history is closer to the goal than the $i^{th}$ history, then we prefer the $j^{th}$ history. This preference labeling scheme operates under the assumption that all actions carry a uniform cost but a non-uniform action cost can also be applied preferring the histories with the lower sum of action costs.

---

**Algorithm 1** Inputs: $\tilde{Q}(h_t, a_{t+1}), P_U(a)$

---
Initialize $\theta$
**while** $\theta$ not converged **do**
    Sample $h_t$ from a set of past search trees
    **for** $k = 1, ..., K$ **do**
        Sample $a_{t+1}^k \sim P_U(a)$
        Compute $\tilde{Q}(h_t, a_{t+1}^k)$
        $\hat{\pi}_{\tilde{Q}}(a_{t+1}^k|h_t) \leftarrow \frac{\exp(\tilde{Q}(h_t, a_{t+1}^k))}{\sum_{j=1}^K \exp(\tilde{Q}(h_t, a_{t+1}^j))}$
    $g_\theta \leftarrow \nabla_\theta \left[ \sum_{k=1}^K \hat{\pi}_{\tilde{Q}}(a_{t+1}^k|h_t) \cdot \mathcal{L}_{\text{ELBO}}(\theta) \right]$
    Update $\theta$ with $g_\theta$

---

We construct the preference dataset, $\mathcal{D}_{pref}$, in which each data point is of the form $(h^i, h^j, y(h^i, h^j))$. This is done by sampling two histories from a search tree, one of which must be a success history, and preference-labeling them. We ensure that $\mathcal{D}_{pref}$ contains an equal number of success-failure and success-success pairs to prevent the significant out-numbering of success-success pairs by success-failure pairs. This happens because the number of success trajectories gathered is greatly out-numbered by failed trajectories (approximately 1/10 in our experiments) especially when using an unguided search. Therefore, if we form $\mathcal{D}_{pref}$ with all potential pairs of histories, we have empirically found that it impairs the learning of the preference between two successful histories.

We train Preference-V, denoted $\tilde{V}$, by setting it as a preference predictor, as in [7, 5, 6]. We approximate the probability of preferring history $h^i$ over $h^j$ as $P[h^i \succ h^j] = \frac{\exp(\tilde{V}(h^i))}{\exp(\tilde{V}(h^i)) + \exp(\tilde{V}(h^j))}$ and learn the function by minimizing the parameters of $\tilde{V}$ with respect to

$$\mathbb{E}_{(h^i, h^j, y) \sim \mathcal{D}_{pref}} [-y(h^i, h^j)[0] \cdot \log P[h^i \succ h^j] - y(h^i, h^j)[1] \cdot \log P[h^j \succ h^i]]$$

where $y(h^i, h^j)[i]$ denotes the $i^{th}$ entry in the preference label. We denote the preference-Q function as $\tilde{Q}$, and train $\tilde{V}$ and $\tilde{Q}$ concurrently by designing the neural network to share the same backbone, with separate heads that accept different inputs as shown in Figure 3.

Given $\tilde{Q}$, we would like to use the energy-based policy of the form $\pi_{\tilde{Q}}(a_{t+1}|h_t) = \frac{\exp(\beta \cdot \tilde{Q}(h_t, a_{t+1}))}{\int_{a'} \exp(\beta \cdot \tilde{Q}(h_t, a'))}$ to guide the search, where $\beta \in \mathbb{R}^+$ is a hyper-parameter that controls the how peaked the distribution is. To sample from this distribution, we must resort to either MCMC sampling or uniform-randomly sampling a fixed number of actions, and then selecting the one with the highest $\tilde{Q}$ value. The former approach is slow yet precise, while the latter is quick but less accurate.

Instead, our strategy is to use VAE to imitate $\pi_{\tilde{Q}}(a_{t+1}|h_t)$. Denote $\mathcal{L}_{\text{ELBO}}$ as the evidence of lower bound (ELBO) loss used to train a VAE whose parameters are denoted as $\theta$. To train our VAE-based policy, denoted $\hat{\pi}_\theta$, it would be ideal to minimize $\mathbb{E}_{a \sim \pi_{\tilde{Q}}(a_{t+1}|h_t)}[\mathcal{L}_{\text{ELBO}}(\hat{\pi}_\theta)]$ with respect to $\theta$. However, sampling from $\pi_{\tilde{Q}}$ is expensive, and the training time for $\hat{\pi}_\theta$ would be too long. So, we use importance sampling to train $\hat{\pi}_\theta$ by sampling actions from a uniform distribution and evaluating the importance weight according to $\pi_{\tilde{Q}}$. Our objective function is $\underset{\theta}{\arg\min} \; \mathbb{E}_{a \sim P_{\text{U}}(a)} \left[ \frac{\pi_{\tilde{Q}}(a|h_t)}{P_{\text{U}}(a)} \cdot \mathcal{L}_{\text{ELBO}}(\theta) \right]$, where $P_U$ denotes the uniform distribution over the action space. Using the gradient of the objective function $g_\theta$, we update the parameters of the policy $\theta$. Intuitively, if an action sampled from the uniform distribution has a high $\tilde{Q}$ value, then it would have a high probability in $\hat{\pi}_\theta$ as well because the ELBO loss for that value would be lower. Unlike raw uniform sampling that does not use VAE, this procedure exposes $\hat{\pi}_\theta$ to a new set of action samples at every gradient step. Algorithm 1 describes the pseudocode for training $\hat{\pi}_\theta$.

## 4  Experiments

We conduct our experiments in three challenging continuous planning domains under partial observability: finding a path in a 2D light-dark room, object fetching with known object classes, and real-world object fetching with unknown object classes. We now describe the POMDP model for each domain.

**2D light-dark room** The objective of the robot is to navigate to a goal position with the minimum number of steps in a 2D plane. A state is the $(x, y)$ location of the robot, an action is $(r, \theta) \in (0, 2) \times [0, 2\pi)$ that defines the movement $(\Delta x, \Delta y) = (r \cos\theta, r \sin\theta)$, the transition model is $(x', y') = (x + \Delta x, y + \Delta y)$, an observation $o \in O$ is $(\hat{x}, \hat{y}) \in \mathbb{R}^2$, $(\hat{x}, \hat{y}) = (x, y) + (\eta_1, \eta_2)$ where $\eta_i \sim \mathcal{N}(0, \sigma(x))$, and $\sigma(x) = 0.01 \cdot (4 - x)^2 + \epsilon$, with $\epsilon = 1e^{-5}$. The robot incurs a reward of -1 for every action and a reward of +100 for reaching the goal region. The discount factor $\gamma$ is set to 1.0, and the maximum planning horizon $H$ is 30. An example problem instance is shown in Figure 4 (a) with the initial robot state shown with a blue dot, the goal region shown with a red circle, and its observations shown in green. A problem instance is defined by an initial state and goal region. The initial state of the robot is sampled from a uniform distribution from the blue box, and the goal region is defined by a circle with a radius of 0.25 whose center is uniform-randomly sampled from the red box.

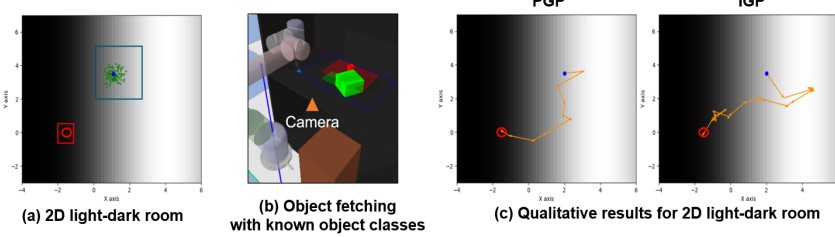

Figure 4: Example problem instances of (a) 2D light-dark room, adapted from [34], (b) Object fetching with known object classes. (c) Example of execution trajectories (orange lines) of PGP and IGP. Note that PGP computes a shorter path.

**Object fetching with known object classes** Figure 4 (b) shows an example problem instance from this domain. The robot's objective is to fetch the target object (red) completely occluded by the non-target object (green) from the cabinet and place it in the goal region (the brown stand to the robot's

right) in a minimum number of steps. The camera is located at the front of the shelf, marked with an orange triangle. A problem instance is defined by the initial poses of the objects, where we first uniform-randomly sample the green box's position from the green transparent box, and then sample the red object location from the region completely occluded by the green box. The orientations are uniformly sampled from $[0, 2\pi)$. The non-target object pose is observable, but not for the target object.

A state is defined as $(q_{\text{robot}}, q_{\text{target}}, q_{\text{non-target}})$, where $q_{\text{robot}}$ is the robot's joint positions and $q_{\text{target}}, q_{\text{non-target}} \in SE(2)$ are the poses of the target and non-target objects, respectively. The action space consists of two action primitives with discrete and continuous parameters, $\text{PICK}(\text{obj}, \mathcal{T}_{\text{pick}})$ and $\text{PLACE}(\text{obj}, \mathcal{T}_{\text{place}})$ where $\text{obj}$ is the object to manipulate and $\mathcal{T}_{\text{pick}}, \mathcal{T}_{\text{place}} \in SE(2)$ are the poses of the gripper. We ensure that each action is feasible using rejection sampling based on the existence of inverse kinematics solution, collision-free motion, and the contact between the object and the gripper, which we describe in detail in the appendix. The transition model is defined with Bullet physics engine [35]. An observation consists of depth and RGB images, denoted $I_{\text{depth}}$ and $I_{\text{RGB}}$ respectively, the union of foreground segmentation mask for each object $Seg_{\text{target}} \cup Seg_{\text{non-target}}$, and the indicator whether the robot is holding an object, $g \in \{0, 1\}$. The observation model is defined as the product of two functions $Z(o|s) = \mathcal{N}(\text{CD}(\text{PCD}(o), \text{PCD}(s))|0, \sigma^2) \cdot \mathbb{1}\{g(o) = g(s)\}$, where CD denotes the Chamfer distance, $\text{PCD}(o)$ and $\text{PCD}(s)$ denote the observed and ground-truth point cloud respectively, $g(o)$ and $g(s)$ are the observation and ground-truth object-holding flag respectively, and $\sigma = 0.0085$. The robot gets +100 if the target object is placed in the goal region, -100 if an object is dropped on the ground, or if the non-target object is placed in the goal region, and -1 for all the other actions. The discount factor $\gamma$ is set to 1.0. The planning depth $H$ is 6.

**Real-world object fetching with unknown object classes** In this domain, the robot operates in the real world and must estimate both the object classes and poses from camera images. Example problems and objects are shown in Figure 1. We implement off-the-shelf perception algorithms and explicitly reason about the uncertainty in the object classifier and pose estimator. More concretely, we model the initial belief as $P(s|o) = P(q, c|o) = P(q|c, o) \cdot P(c|o)$, where $c$ is the class label and $q$ is SE(3) pose of the object. We use Mask R-CNN [36] with dropout sampling [37] to model $P(c|o)$, and a PointNet [38]-based architecture for the pose estimator which also gives a distribution over the potential poses. The details of the perception system is in the appendix. A state is defined as $(q_{\text{robot}}, q_{\text{target}}, q_{\text{non-target}}, c_{\text{target}}, c_{\text{non-target}})$, where $c_{\text{target}}, c_{\text{non-target}}$ are the class labels of the target and non-target objects. Other quantities in the POMDP are equivalent to the previous domain.

We have two hypotheses. First, for learning from search trees, preference-based learning is more data efficient than imitation learning, and second, guided search is faster than an unguided search. To validate these, we compare PGP and its variant, success-fail-preference-guided POMCPOW (SF-PGP), where the value and policy networks are trained to prefer the success trajectory over a failure trajectory but not necessarily to prefer shorter success trajectories, to following methods. The first is **unguided POMCPOW** (UNGUIDED) which is a standard POMCPOW that samples actions with a uniform random policy and evaluates a leaf node using a rollout. The second is **imitation-guided POMCPOW** (IGP) which uses an imitation policy and MSE-V function for guiding POMCPOW. The imitation policy is trained to imitate actions on success histories, and MSE-V is trained to predict the future sum of rewards of the given trajectory using a mean squared error loss. For the guided search, we replace the rollout with value function evaluation, and the uniform sampling over actions using the trained policy. For fetching domains, we learn the policy for just the place action.

Figure 5 (a) shows the result for the light-dark room domain. In the first two columns, preference-based approaches, PGP and SF-PGP, achieve significantly higher success rates than IGP when they are trained with the same number of tree searches and computes, supporting our hypotheses. With 10 search trees and 100 simulations, PGP and SF-PGP achieve 80% success rate, while IGP and UNGUIDED require 200 simulations to achieve that. With 100 search trees, IGP gets better, but the IGP and SF-PGP are more efficient and achieve higher success rates. The third and fourth columns show that PGP and SF-PGP compute shorter successful trajectories more efficiently compared to UNGUIDED and IGP when using 10 and 100 tree searches, although UNGUIDED eventually gets similar

optimality as PGP and SF-PGP. Figure 4 (c) shows the comparison of the trajectories computed by PGP and IGP.

Figure 5 (b) shows the result for the fetching domain with known object classes. In the first column, we again can see that PGP and SF-PGP achieve higher success rates than IGP and UNGUIDED when using the same number of simulations and using 10 tree searches. When using 100 tree searches (second column), we see that IGP improves, but PGP and SF-PGP still outperforms it, demonstrating both computational and data efficiency. Notably, when using just 10 search trees, PGP outperforms SF-PGP, unlike the previous domain. This is because, in the previous domain, the length of the trajectory does not influence success or failure as the planning horizon is long and the robot has ample opportunity to correct its behavior. In contrast, in this domain, shorter trajectories tend to have higher success rates because the robot needs to fetch the target object within 6 steps, otherwise, it is a failure. This indicates that the success-success comparison scheme in PGP is helpful in this domain. In the third and fourth columns, we see that PGP outperforms all other algorithms in terms of optimality. These results again support our hypothesis.

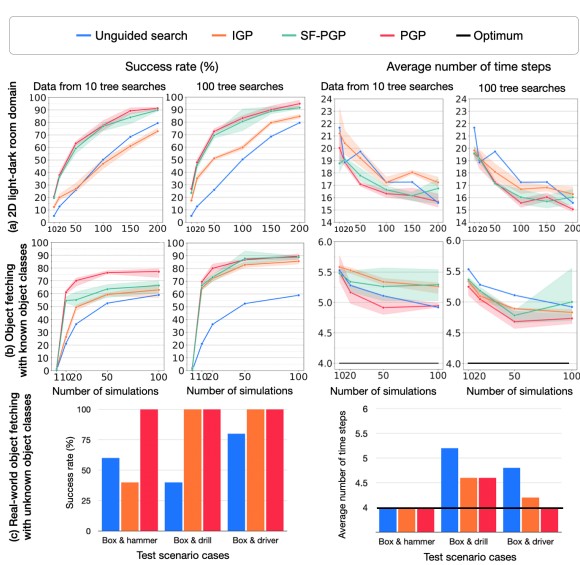

Figure 5: Success rates vs a number of simulations for 10 and 100 tree searches (columns 1 and 2 respectively), and an average number of steps to the goal for successful plans for 10 and 100 tree searches (columns 3 and 4 respectively) for (a) light-dark room domain and (b) object fetching with known object classes. (c) Success rates (column 1) and number of steps to the goal (column 2) in real-world object fetching using 50 simulations and 100 tree searches.

Figure 5 (c) shows the results from the real robot experiments from the scenario shown in Figure 1 for target objects from different classes. From the first column, we see that PGP again outperforms UNGUIDED in all objects in terms of success rate, and outperforms IGP in the hammer case. In two other cases, IGP gets the same success rate as PGP. The second column shows that for success trajectories, PGP and SF-PGP find the most optimal solution among all the baselines. We include a more exhaustive comparison in the simulated version of this domain for all baselines in the appendix.

## 5 Limitations and future directions

In this work, we propose a preference-based learning algorithm for guiding POMCPOW and show that it is more computationally efficient than UNGUIDED and more data-efficient than IGP in several robotics domains. One key limitation of our framework, as with any POMDP-based planner, is the heavy reliance on the quality of the perception algorithm (more exhaustive limitations are included in the appendix). To focus on planning, we deliberately simplified the perception problem in our real-robot domain by considering a limited set of objects. Even with this simplification, however, we had many cases where we lacked particles that matched the real world, and often had geometrically infeasible object poses and shapes where objects penetrated each other. We used particle re-invigoration and re-planning, which helped, but they incurred additional computation costs. These findings suggest that the learning-to-guide-planning paradigm shows promise in reducing planning time. However, they also highlight the need for significant advancements in representation and uncertainty quantification for perception to improve the initial belief generator.

**Acknowledgments**

This work was supported by Institute of Information & communications Technology Planning & Evaluation (IITP) grant funded by the Korea government(MSIT) (No.2019-0-00075, Artificial Intelligence Graduate School Program(KAIST)), (No.2022-0-00311, Development of Goal-Oriented Reinforcement Learning Techniques for Contact-Rich Robotic Manipulation of Everyday Objects), (No. 2022-0-00612, Geometric and Physical Commonsense Reasoning based Behavior Intelligence for Embodied AI), and Samsung Electronics.

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

# Appendix

## A. Limitations

**Observation model** Our algorithm can show differences in performance depending on how we set up the observation model. Our algorithm approximates the belief through a particle filter, and the weight of the particles is proportional to the observation likelihood. If an inappropriate observation model is set, the updated belief will not properly reflect this information even if the information gathering action is taken. Most failures in the fetching domain occurred because there was no particle similar to the true state. For example, We set the Gaussian model using Chamfer distance as the observation model in fetching domains, as described in Section 4. This model can sometimes be calculated to have a small distance between the actual and other class objects when parts of objects in different classes are similar to each other. This was the main reason for the task to fail when we set the number of particles to small. You could try to deal with this problem by increasing the number of particles, but this is not practical because it increases planning time. Therefore, in order for the algorithm to work effectively with a small particle, it is necessary to carefully set the observation space and observation model so that the observation obtained through the information gathering action can effectively reflect information on the true state.

**Belief generator** Our algorithm is difficult to deal with unseen objects in fetching domain due to the limitations of the belief generator. As far as we know, there is no perception algorithm yet that can predict the shape and pose of all objects present in the scene through one image obtained from a fixed view and propose particles with reflecting uncertainty by occlusion. Therefore, we limited the classes of possible objects in the domain and assumed that if we could know the class of each object, we could know the shape of the object, in order to construct a generating pipeline that can operate by combining existing algorithms, such as Mask R-CNN [1] with dropout sampling [2] and PointNet [3]-based pose estimator.

**Planning time** As you can see in the supplementary video, our algorithm can make a plan with less time than unguided search. However, it is not enough to apply it as a real-time yet. We have tested our algorithm with CPU of AMD Ryzen 5600x and GPU of NVIDIA RTX 3070, and the average planning time for 50 simulations are shown in Table 1. It performed fairly well with a small number of particles through guided search, but more particles are needed for a higher success rate. Our algorithm based on serial tree search has a trade-off between planning time and performance improvement due to an increase in the number of particles. In [4], they proposed an algorithm that parallelly performs MCTS. In [5], they proposed an algorithm that operates at a real-time level in combination with this algorithm and a physical simulator that is advantageous for parallel processing. If we can also adapt these methods to tree search in parallel, we expect to effectively reduce the planning time of our algorithm.

|          | Box & hammer | Box & drill | Box & driver |
|----------|--------------|-------------|--------------|
| Unguided | 142.46       | 136.60      | 149.69       |
| IGP      | 75.07        | 74.96       | 70.57        |
| SF-PGP   | 65.67        | 72.64       | 75.60        |
| PGP      | 55.07        | 64.49       | 62.43        |

Table 1: Planning time of the algorithms with 50 simulations measured in seconds. The test was done with CPU of AMD Ryzen 5600x and GPU of NVIDIA RTX 3070. Each row represents different algorithms whil each column shows different scenarios. Overall, the guided searches are faster than unguided searches, yet they are not enough for real-time performance.

# B. Implementation details for training

## B.1. Architecture of networks

**Backbone transformer**

- Light-dark room domain
    - Number of layers: 3
    - Number of attention heads: 1
    - Embedding dimension: 128
    - Input encoding
        * Observation input dimension: 2
        * Action input dimension: 2
    - Nonlinearity function: GeLU
    - Sequence length: 61
- Fetching domains
    - Number of layers: 3
    - Number of attention heads: 2
    - Embedding dimension: 256
    - Input encoding
        * Observation input dimension:
            · RGB-D: $64 \times 4$
            · Grasp detection identifier: 1
        * Image encoder:
            · Number of convolutional layers: 4
            · Number of max pool layers: 4
            · Kernel size: 3
            · Stride: 1
            · Dimension of feature maps: (3, 16, 32, 64, 128)
        * Action input dimension: 8
    - Nonlinearity function: GeLU
    - Sequence length: 13

**Header of value networks**

- Light-dark domain
    - Input dimension: 128
    - Output dimension: 1
    - Number of layers: 1
- Fetching domains
    - Input dimension: 256
    - Output dimension: 1
    - Number of layers: 2
    - Hidden dimension: 256

**Header of policy network**

- Light-dark domain
    - Input dimension: 2

- – Output dimension: 2
- – Condition dimension: 128
- – Latent dimension: 64
- – Number of encoder layers: 3
- – Number of decoder layers: 3
- – Beta
  - ∗ Light-dark
    - · Imitation: 0.25
    - · Preference: 0.5
  - ∗ Fetching domains
    - · Imitation: 0.25
    - · Preference: 0.5
- • Fetching domain
  - – Input dimension: 3
  - – Output dimension: 3
  - – Condition dimension: 256
  - – Latent dimension: 64
  - – Number of encoder layers: 4
  - – Number of decoder layers: 4
  - – Beta
    - ∗ Light-dark
      - · Imitation: 0.25
      - · Preference: 0.5
    - ∗ Fetching domains
      - · Imitation: 0.25
      - · Preference: 0.5

## B.2. Hyperparameters

- • Light-dark domain
  - – Batch size
    - ∗ Value
      - · 128 for the dataset with 10 search trees
      - · 512 for the dataset with 50 search trees
      - · 2048 for the dataset with 100 or larger number of search trees.
    - ∗ Policy
      - · 64
  - – Learning rate: 1e-4 (1e-5 for dataset with 10 search trees)
  - – Learning rate scheduler: None
  - – Optimizer: AdamW
  - – Weight decay: 1e-5
  - – Dropout: 0.1
- • Fetching domains
  - – Batch size
    - ∗ Value
      - · 128 for the dataset with 10 search trees
      - · 512 for the dataset with 50 search trees
      - · 2048 for the dataset with 100 or larger number of search trees.

* Policy
  · 64

- Learning rate: 0.0001 (0.00001 for dataset with 10 search trees)

- Learning rate scheduler: Multi-step scheduler with gamma = 0.1 and milestone=4000

- Optimizer: Adam

- Dropout: 0.1

## C. Guiding POMCPOW with the learned functions

Using $\tilde{V}, \tilde{Q}$, and $\hat{\pi}_\theta$, we guide POMCPOW in order to speed up planning. Algorithm 1 shows the pseudocode for the guided POMCPOW. The guided search takes in the trained value function and policy, initial belief, the total number of simulations $n$, and the maximum planning horizon $T$. The variables used throughout the algorithms are: an execution history $h_t = (o_0, a_1, o_1, ..., a_t, o_t)$, a list of child nodes $C$, a number of visits to a node $N$, a number of times that a given observation node has been generated $M$, and $B$ and $W$ are the list of the list of belief states and the weight associated with it respectively. $C, N, M, B, W$ are implicitly initialized to $\emptyset$ or 0.

Procedure GUIDEDSEARCH in Algorithm 1 describes the overall process of guided POMCPOW. The procedure takes initial belief $b_0(s)$, $\tilde{V}$, $\hat{\pi}_\theta$, $n$, $T$. For each iteration, we sample a particle $s$ from $b_0(s)$, then run SIMULATE procedure. iterate through the particles by simulating with each of them, and outputs action $a$ with the highest Q value backed up by the simulation.

The new action is sampled with a procedure ACTIONPROGWIDEN similar to the one proposed in POMCPOW[6]. However, to efficiently sample an action rather than resorting to a random policy, we use $\hat{\pi}_\theta$ to produce a new action sample (9). Then, an action is selected according to UCB1 (11).

The SIMULATE is a recursive function that terminates when it reaches the maximum search depth of 0. Otherwise, it samples an action with ACTIONPROGWIDEN, expands observation node (line 6 to 11) and action node (line 12 to 18). However, guided POMCPOW differs in that it leverages $\tilde{V}$ instead of rollout by random policy (line 14) to reduce planning time.

There are several hyperparameters for POMCPOW: progressive widening constants $(k_a\alpha_a, k_o, \alpha_o)$ and exploration constant $c$ for UCB. We set the progressive widening parameter as $(k_a\alpha_a, k_o, \alpha_o) = (0.5, 0.5, 0.5, 0.5)$ for 2D light-dark room domain and $(k_a\alpha_a, k_o, \alpha_o) = (3.0, 0.15, 3.0, 0.15)$ for two object fetching domains. Since the value outputs of each method in each domain have different scales, we set the exploration constant $c$ differently for each method. In 2D light-dark room domain, we set it as 20 for PGP and SF-PGP, and 50 for IGP and unguided search. In object fetching domain with known object class, we set it as 0.5 for PGP and SF-PGP, and 100 for IGP and unguided search. In object fetching domain with unknown object class, we set it as 20 for PGP and SF-PGP, and 200 for IGP and unguided search.

### D. Experiment setup for fetching domains

To ensure that each pick or place actions are in contact with the object, we use rejection sampling. To sample PICK, we first choose a particle from a belief state, and for the given object, we choose a point on the upper surface of the object as a contact point. Then, we check two things: the *cosine similarity* between the normal vector at the contact point and the z-axis must be within 0.95, and the entire suction cup must be enclosed inside the object when we project it down along the z-axis. If these are not met, we sample another PICK and repeat the procedure. For PICK that passed the tests, we check the existence of an IK solution and collision-free motion plan using biRRT [7], and if they do not exist, we sample another PICK and restart from the beginning. PLACE simply samples a pose that is either inside the cabinet or the goal region.

---

**Algorithm 1** Guided POMCPOW

---

1: **procedure** GUIDEDSEARCH($b_0(s)$, $\tilde{V}$, $\hat{\pi}_\theta$, $n$, $T$)
2:     $Q \leftarrow -\infty, h \leftarrow \emptyset$
3:     **for** $i \leftarrow 0$ to $n$ **do**
4:         $s \sim b_0(s)$
5:         SIMULATE($s, h, T, \hat{\pi}_\theta, \tilde{V}$)
6:     **return** $\underset{a}{\arg\max}\, Q(ba)$

7: **procedure** ACTIONPROGWIDEN($h$)
8:     **if** $|C(h)| \leq k_a N(h)^{\alpha_a}$ **then**
9:         $a \sim \hat{\pi}_\theta(h)$
10:        $C(h) \leftarrow C(h) \cup \{a\}$
11:     **return** $\underset{a \in C(h)}{\arg\max}\, Q(ha) + c\sqrt{\frac{\log N(h)}{N(ha)}}$

1: **procedure** SIMULATE($s, h, d, \hat{\pi}_\theta, \tilde{V}$)
2:     **if** $d = 0$ **then**
3:         **return** $0$
4:     $a \leftarrow$ ACTIONPROGWIDEN($h, \hat{\pi}_\theta$)
5:     $s', o, r \leftarrow G(s, a)$
6:     **if** $|C(h)| \leq k_a N(h)^{\alpha_a}$ **then**
7:         $M(hao) \leftarrow M(hao) + 1$
8:     **else**
9:         $o \leftarrow$ select $o \in C(ha)$ w.p. $\frac{M(hao)}{\sum_o M(hao)}$
10:     append $s'$ to $B(hao)$
11:     append $\mathcal{Z}(o|\, s, a, s')$ to $W(hao)$
12:     **if** $o \notin C(ha)$ **then**
13:         $C(ha) \leftarrow C(ha) \cup \{o\}$
14:         $total \leftarrow r + \gamma \tilde{V}(hao)$
15:     **else**
16:         $s' \leftarrow$ select $B(hao)[i]$ w.p. $\frac{W(hao)[i]}{\sum_{j=1}^m W(hao)[j]}$
17:         $r \leftarrow R(s, a, s')$
18:         $total \leftarrow r + \gamma$SIMULATE($s', hao, d - 1$)
19:     $N(h) \leftarrow N(h) + 1$
20:     $N(ha) \leftarrow N(ha) + 1$
21:     $Q(ha) \leftarrow Q(ha) + \frac{total - Q(ha)}{N(ha)}$
22:     **return** $total$

---

### Perception System

In real-world object fetching with unknown object classes domain, we use Mask R-CNN [1] with dropout sampling [2] to model the class distribution of an object, $p(c|o)$. For efficient data procurement, We utilize the Bullet physics simulator [8] to collect RGB images and mask annotation. Nonetheless, the data from the simulation, especially the RGB image, does not generalize to the real-world scene due to the sim2real gap. To overcome this, we render the image with NVISII [9] to create photo-realistic texture while using the mask annotation from Bullet simulator and train Mask R-CNN with these images.

To estimate $p(q|c, o)$, we train a pose estimator per each class. It is a PointNet [3]-based architecture with 2 heads of MLP layers for orientation and position. We bin the orientation into 2048 classes and train the orientation head to predict the unit quaternion closest to the ground truth orientation while the position head is trained with regression. During an inference, we decide which pose estimator to use according to the class label predicted by Mask R-CNN. Then, we feed the partial point cloud obtained from the depth image and the segmentation result and get $p(q|c, o)$. To generate the data, we randomly sample about 130k positions on the cabinet and yaw angle (roll and pitch angles of the objects are fixed to 0). Then, we load random objects to the Bullet simulator with the sampled pose, capture the partial point cloud using the depth camera, and label them with corresponding object poses given by the simulator.

### Initial belief generator

In real-world object fetching with unknown object classes domain, the initial belief generator uses the result from the perception system to sample initial belief over the state $p(s|o)$. For class uncertainty, we sample $N_{shape}$ number of class labels from the class distribution output by Mask R-CNN. Then, for each class, we run our pose estimator to sample $N_{pose}$ number of poses. There are total

$N_{shape} \times N_{pose}$ candidates for beliefs, and we randomly choose $N_{particle}$ particles for the initial belief with rejection sampling. The criteria for rejection is the existence of collision between objects or cabinet, grasp affordance of the vacuum gripper, and the misclassification of target(red) objects (i.e. hammer, drill, driver) to the non-target object class label (i.e. box) or vice-versa.

## E. Evaluation

### Qualitative results for object fetching with known object classes

Figure 1 (a) shows the visualization of the outputs of value networks, which are normalized between 0 and 1, for PLACE action on the same scene. After the robot picks the non-target object, to achieve information-gathering actions, the robot should prefer the region where the target object can be visible, such as the side of the area, except the area where the target object can be invisible. Our approach can tell the clear difference between such areas, whereas other approaches do not show clear differences, or even show the same value across all areas. (b) and (c) from Figure 1 shows the action samples when PICK on both target and non-target object. When a target object is picked, the robot should prefer the action samples around the goal region. Here, our model shows a high preference in that area. When the non-target object is picked, the robot should prefer the action samples where the target object can be visible, and also not the goal area. Our approach avoids around the target object, whereas others take action samples around the target object and goal area.

### Quantitative results of fetching domains with different sizes of data

We compared the results of a guided search using networks trained in various data sizes to evaluate data efficiency. Figure 2 (a) shows the result of object fetching with known object classes domain, using data from 50 and 300 search trees in addition to the results shown in the main document.

The first row shows that the results are consistent with our hypotheses. With 50 and 300 tree searches, SF-PGP and PGP show higher success rates than IGP. The second row shows that PGP achieves a more optimal solution than other approaches. With 300 tree searches, PGP finds near-optimal plans, while other approaches struggle to do so. IGP comes close, but note that it has a lower success rate than PGP. Figure 2 (b) shows the result of real-world object fetching with unknown object classes domain evaluated in PyBullet [8] simulation, using data from 10 and 100 search trees. The two plots on the left show that the results are consistent with our hypotheses. With 10 and 100 tree searches, SF-PGP and PGP show higher success rates than IGP. The two plots on the right show that PGP finds the most optimal plans.

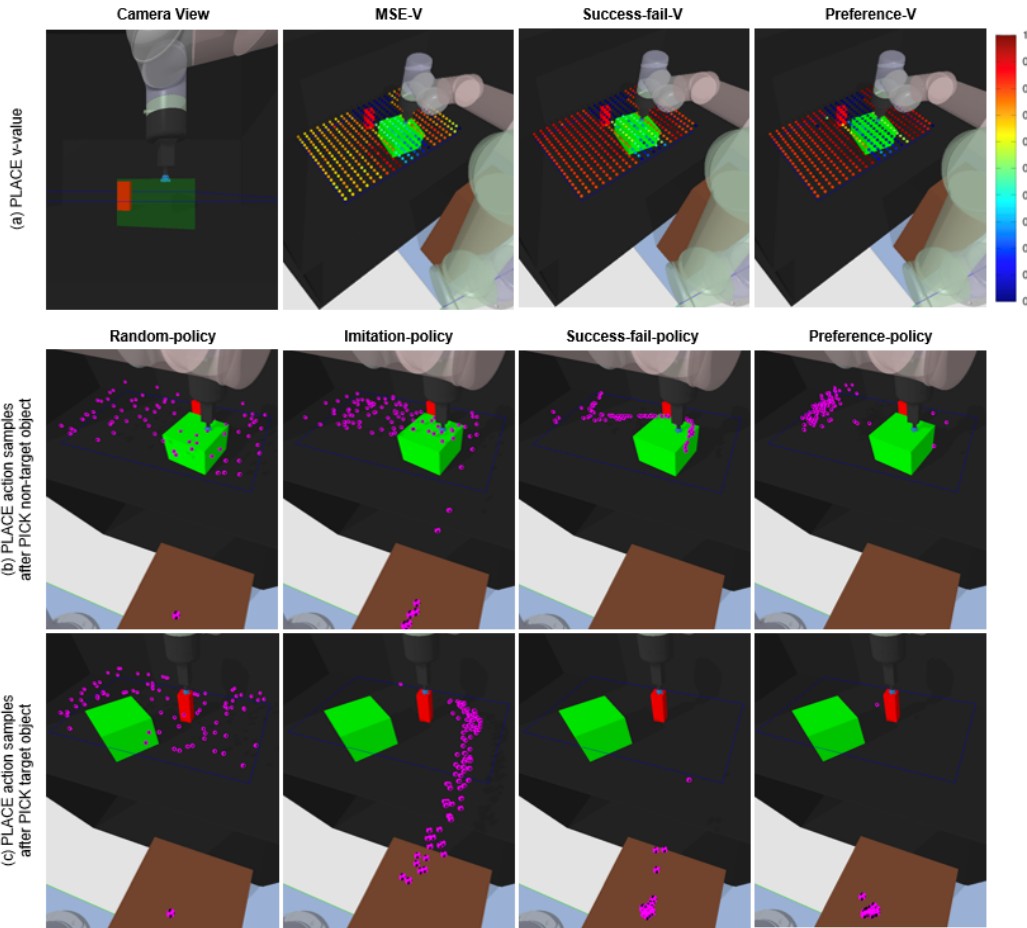

Figure 1: **Qualitative comparisons on object fetching with known object classes.** (a) shows normalized V-values of PLACE actions obtained from 3 different value networks (b) purple dots show 100 PLACE action samples from the trained policy after PICK non-target object. (c) purple dots show 100 PLACE action samples from the trained policy after PICK target object when the target object is visible. The networks used in these figures were trained using data from 50 search trees.

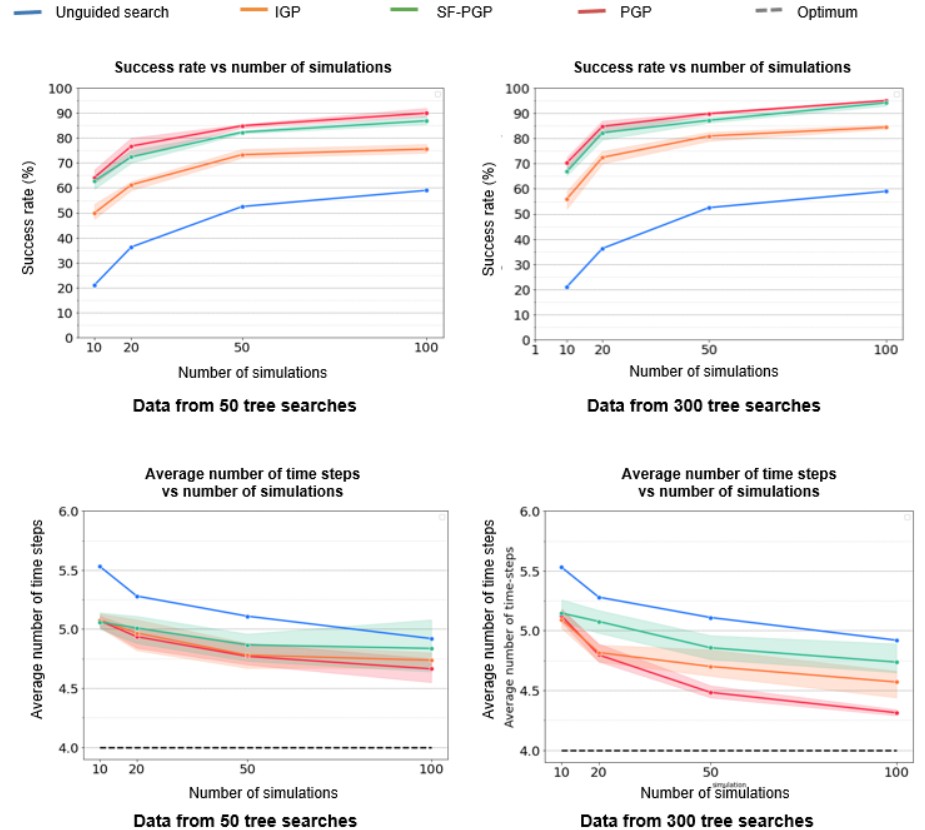

(a) **Quantitative results of object fetching with known object classes.** Data from 50 search trees and 300 search trees are compared. The first row shows the success rates of guided search using networks learned from data obtained from the different number of search trees. The second row shows the average time-step of success trajectories obtained from these searches.

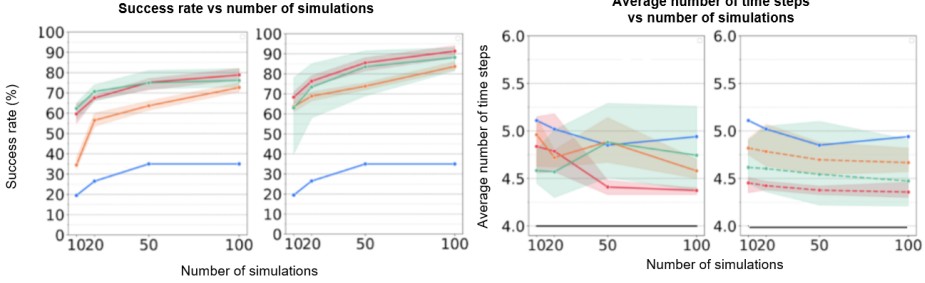

(b) **Quantitative results of real-world object fetching with unknown object classes in PyBullet [8] simulation.** Data from 10 search trees and 100 search trees are compared. The two plots on the left show the success rates of guided search using networks learned from data obtained from the different number of search trees. The two plots on the right show the average time-step of success trajectories obtained from these searches.

Figure 2: **Quantitative comparisons on object fetching domains** All plots depict the mean and 95% confidence intervals (CIs) based on 400 experiments conducted with trained networks using three different random seeds.

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
