# OpenReview forum: "Preference learning for guiding the tree search in continuous POMDPs"
_robot-learning.org/CoRL/2023/Conference — CoRL 2023 Poster_

### Official Review · Reviewer_TVga · 2023-07-17

**Confidence:** 5
**Originality:** Very Good
**Technical Quality:** Very Good
**Clarity Of Presentation:** Very Good
**Impact:** 4

**Recommendation:**

Strong Accept: I recommend accepting the paper and will argue for my recommendation even if other reviewers hold a different opinion.

**Review:**

This is a very nice paper, both clearly written and well motivated. The observation that the ranking between histories (instead of computing their actual values) is sufficient to efficiently guide the tree search makes sense and is effectively exploited in PGP. The method itself is sound and described with sufficient detail. The experiments are generally carried out well and demonstrate the effectiveness of PGP (including on a real-world object fetch scenario).

One limitation of PGP that wasn't mentioned in the paper is that the current history ranking scheme assumes that the problem includes a set of goal states and that histories are labelled according to whether they reach a goal state or not. However, in many POMDPs we don't have any goal states, or the information whether a history has reached a goal state can not be inferred from the history itself (due to the partial observability of states). The paper should make it clearer what the assumptions are (existence and identifiability of goal state) or whether the method can be extended to the general POMDP case.

Some minor comments:
- g_{\theta} in Algorithm 1 is not defined.

- In the videos it is not clear what's happening in the upper-left corner. Is this some representation of the beliefs?

**Quality Of The Limitations Section:**

Additional details required

**Questions For Rebuttal:**

- Section 3: "This preference labelling scheme operates under the assumption that all actions carry a uniform cost." => This needs to be explained further. Isn't that a very strong assumption?

- During deployment, the learned policy is only used to sample new actions, in case PW decides to add an action to a history node. Previous works (e.g. Alpha-Go or Lets-Drive) have additionally used the policy to guide the action selection strategy during the tree search. Is there a reason why PGP doesn't do that?

- The current architectures of the value and policy networks require a goal state as an input, but it is not clear why this is needed. Wouldn't a simple binary value "goal reached" and "goal not reached" suffice? As mentioned above, for problems with goal states, we don't necessarily know the exact goal state reached by a history.

**Robotics Focus:**

Sufficient demonstration on hardware

**Summary Of Paper:**

This paper proposes Preference-Guided POMCPOW (PGP), and extension of POMCPOW that learns a policy and a value function to guide the tree search. In contrast to existing works, PGP aims to reduce the cost and data requirements needed to learn a policy and a value function that is used to guide POMCPOW's Monte Carlo Tree Search. At its core, PGP learns a value function and a policy (represented by transformer and VAE based networks) using preference learning, by labelling histories obtained from previous simulations according to whether they were successful in reaching the goal. The premise is that preference learning significantly reduces the data requirements to learn useful policies and value function compared to previous methods. PGP is tested on a set of POMDP problems where it shows superior performance compared to unguided POMCPOW and imitation-guided POMCPOW.

**Summary Of Recommendation:**

The paper proposes an interesting and novel extension of POMCPOW and its efficiency is demonstrated on a set of POMDP problems, including a real-world object fetch scenario. While in some parts the paper requires further clarifications (see comments and questions above), I believe that its contributions are substantial enough to warrant publication.

---

### Official Review · Reviewer_dipf · 2023-07-18

**Confidence:** 4
**Originality:** Good
**Technical Quality:** Fair
**Clarity Of Presentation:** Very Good
**Impact:** 3

**Recommendation:**

Weak Accept: I recommend accepting the paper, but will not argue for my recommendation if the majority of other reviewers have a different opinion.

**Review:**

POST-REBUTTAL UPDATE: The authors addressed my major comment (#1 below) empirically and I updated my recommendation as weak accept.


There are two major flaws of the paper:

1) The main motivation for using preference feedback is that it is easier for humans to provide (than actual scalar values). However in this work, the reward function is already known and it is possible to do standard regression to learn the value function instead of preference-based learning. The paper claims preference feedback "can create much larger data from a single search tree than the one that regresses the value from history". While it is true that preference dataset will be larger, it is not true that this dataset will be more informative. If that was the case, one could go further and create rankings of three trajectories to get better datasets for learning the value functions. Data processing inequality tells us that if X implies Y which implies Z, then Z cannot give more information about X than Y. In this case, true value function (X) implies the true values of the realized trajectories (Y), and these values (Y) imply the preferences between them (Z). Therefore, information theoretically, the preference dataset cannot be more informative, albeit larger, than a dataset that consists of (trajectory, value) pairs.

2) The work cites some recent LLM works for preference-based learning, which is fine. But it does not give enough credit to the large body of preference-based learning in robotics. Of course, I acknowledge the fact that citations [5-7] are either in robotics or are closely related, but  there have been a lot of work in robot learning that developed different forms of preference feedback and analyzed them information theoretically. I would suggest the authors to dive deeper into that literature.

3) Minor: Bibliography information is missing on reference 17.

**Quality Of The Limitations Section:**

Additional details required

**Questions For Rebuttal:**

The first point I raised in my review is my biggest concern about the paper. If authors can theoretically prove me wrong or at least do an ablation study that shows preference dataset is better, I will be happy to recommend the acceptance of the paper.

**Robotics Focus:**

Sufficient demonstration on hardware

**Summary Of Paper:**

This paper proposes a method to solve POMDPs with continuous action spaces. The idea is to guide the tree search to solve the POMDP using preference-based feedback where successful trajectories are preferred over unsuccessful ones, and shorter successful trajectories are preferred over longer successful trajectories. The approach is validated with a few simulated experiments and in a real-robot experiment.

**Summary Of Recommendation:**

This paper uses preference feedback for a task which I don't believe requires preference-feedback. The knowledge of the reward function should make preference-based learning unnecessary, and I expect a simpler regression approach to work not worse than the preference-based method.

POST-REBUTTAL UPDATE: The authors conducted an additional study that empirically shows the benefit of preference feedback over value feedback, even though this is information theoretically counter-intuitive. I updated my recommendation from weak reject to weak accept.

---

### Official Review · Reviewer_kYWh · 2023-07-19

**Confidence:** 3
**Originality:** Fair
**Technical Quality:** Good
**Clarity Of Presentation:** Good
**Impact:** 3

**Recommendation:**

Weak Accept: I recommend accepting the paper, but will not argue for my recommendation if the majority of other reviewers have a different opinion.

**Review:**

Strengths:

Originality: The paper proposes an original framework that combines preference learning with tree search in continuous POMDPs, addressing the challenges in complex real-world scenarios.

Quality: The effectiveness of the proposed framework is supported by representative experiments in both simulated and real-world environments. The comparison with suitable baselines demonstrates the advantages in terms of effectiveness and efficiency.

Clarity: The paper provides a fair introduction to the problem and the proposed framework. The figures and discussions enhance the clarity of the presented concepts and results to some degree.

Significance: The paper contributes to the study of planning approaches in partially observable continuous domains, offering a somewhat practical framework for real-world applications with improved data and time efficiency.


Weaknesses:

- Dependence on observation model: The performance of the framework heavily relies on the quality of the observation model without any extra layer of representations, which can be a limitation in terms of perception and scene understanding.

- Applicability to real-world tasks: While the proposed framework is more time-efficient than existing baselines, it seems to still falls short of real-time operation, which limits its practicality for many applications.

- Those two points above are acknowledged by the authors, however, the paper does not discuss these issues nor possible solutions in detail.

- Limited variety of real-world experiment setup: The set of object classes and environmental setups in the real-world experiments do not cover a wide enough range of scenarios, raising concerns about the generalizability of the approach.

- Assumption of uniform action cost: The assumption of uniform action cost may not hold in many real-world problems, which contradicts the claim of increased practicality compared to existing methods.

Minor issues:
- Style issues in figures: Some figures, such as Figure 5, suffer from small font size and readability issues, affecting the overall presentation of the experiment results.

- Unclear discussion: At line 286, authors mention that PGP finds the most optimal solution among all the baselines. However, since there is an explicit distinction made between PGP and SF-PGP, this statement becomes unclear, since in some experiments SF-PGP outperforms PGP. It would be better to replace “PGP” with “PGP and SF-PGP” or “PGP-based”. So, a clarification or rephrasing is needed to avoid confusion.


**Quality Of The Limitations Section:**

Additional details required

**Questions For Rebuttal:**

- The experiments show examples of settings with a single occluding object that the robot is capable of moving; then, the robot moves the occluding object out of the way and picks up the target afterwards. How might planning change in other settings where the robot cannot move the occluding object, or if there are multiple occluding objects? It would be great if you could extend this analysis (surely acknowledging the limited time).

- Is it possible to extend the preference labeling scheme for actions carrying non-uniform cost?

- Can you provide details about the runtimes of those algorithms? Neither the paper nor the videos are informative.

- The task covered is simple, with two objects, and requiring a couple of steps. What’s the scalability of this approach?

- The accompanying video could be enhanced to better explain the steps of the process.


**Robotics Focus:**

Sufficient demonstration on hardware

**Summary Of Paper:**

The paper introduces PGP (Preference-Guided POMCPOW), a new framework for planning tasks in partially observable, continuous domains. As the main contribution, PGP improves upon POMCPOW by incorporating preference learning to guide the tree search. The authors generate a preference dataset by ranking trajectories based on reaching the goal state. They then use this dataset to train value and action-value functions using a transformer. They propose an energy-based method for policy learning and employ a variational autoencoder (VAE) to approximate it.
The authors evaluate PGP in various environments, including both simulated and real-world scenarios. They compare its performance against baselines such as unguided and imitation-guided POMCPOW. The results demonstrate that PGP, in general, achieves higher success rates and improved efficiency in the tested environments. Although the scenarios do not cover a wide range, the findings highlight the practicality of PGP for continuous POMDPs.


**Summary Of Recommendation:**

The paper introduces a novel method for preference-based dataset generation and improves upon existing approaches for continuous POMDP with preference learning. While the proposed method demonstrates increased data and time efficiency, its real-time applicability remains limited. The experiments conducted in simulation and real-world settings validate the effectiveness of the approach. Overall, the paper presents a strong combination of learning approaches in perception and planning to address a challenging task in a partially observable environment.

---

### Official Review · Reviewer_8hjC · 2023-07-19

**Confidence:** 3
**Originality:** Good
**Technical Quality:** Good
**Clarity Of Presentation:** Good
**Impact:** 3

**Recommendation:**

Weak Accept: I recommend accepting the paper, but will not argue for my recommendation if the majority of other reviewers have a different opinion.

**Review:**

The paper is well written, addressing an interesting and challenging problem and convincing empirical results. The review of related work is quite extensive and convincing, limitations are discussed in great detail. While the paper is overall convincing, I have a few comments, see below.

**Quality Of The Limitations Section:**

Limitations are addressed clearly

**Questions For Rebuttal:**

1.	Introduction: There is no clear statement of contribution, and no clear problem statement. While the introduction describes the problem setup, it would be good to add the latter in clear mathematical terms.

2.	Introduction line 75: Why is a multi-modal policy needed? There should be at least some high-level explanation that does not require reading [17].

3.	Approach: The definition of T_i  is not clear, please clarify.

4.	Results: Figure 5 is quite dense, it might be good to find a little bit of extra space throughout the paper to subcaptions below each row to improve readability. Further, the third row misses y-labels, it is not immediately clear what is shown. Also, optimum appears only in the second third row, but not in the first – why is that?


**Robotics Focus:**

Sufficient demonstration on hardware

**Summary Of Paper:**

Review for “Preference learning for guiding the tree search in continuous POMDPs“

The paper studies planning problems in partially observable environments. A common approach is using POMDPs, yet they are not applicable for continuous space problems. POMCPOWs present an alternative, yet are challenged in high-dimensions. The authors, present a framework to guide POMCPOW using pair-wise preferences based learning. Preferences are provide by automatically analyzing past search trees: a successful trajectory is to be preferred over a non-successful one, and between two successful trajectories preference is given to the shorter one.
This pairing of search tree histories allows for efficiently generating large data sets. Using reward learning allows for obtaining successful reward functions. The framework is validated in two simulation and one real-world experiments, showing its advantages over the state-of-the-art with respect to success rate and efficiency of the found solutions.


**Summary Of Recommendation:**

Overall, this is a well written and paper, relevant to the CoRL community. However, there are a few minor issues that should be addressed.

---

### Decision · Program_Chairs · 2023-08-30

**Decision:**

Accept (Poster)

**Comment:**

The paper proposes to use preference guiding in POMCPOW to improve performance where POMCPOW is a tree-search based algorithm for solving continuous valued POMDP problems. The proposed approach learns a value function and policy using preference learning to guide the tree search. In experimental evaluation, the proposed approach outperforms comparison methods. The paper is well written and motivated. Using ranks instead of actual values is a valuable idea.